**Investigation**

# Locally adaptive inversions in structured populations

Carl Mackintosh [1,2,3,4] Michael F Scott [5] Max Reuter [1] Andrew Pomiankowski [1,2,*]

[1]Department of Genetics, Evolution, and Environment, University College London, Gower Street, London WC1E 6BT, UK
[2]CoMPLEX, University College London, Gower Street, London WC1E 6BT, UK
[3]CNRS, UMR7144 Adaptation et Diversité en Milieu Marin, Station Biologique de Roscoff, Roscoff 29680, France
[4]Sorbonne Universités, UPMC Université Paris VI, Roscoff 29680, France
[5]School of Biological Sciences, University of East Anglia, Norwich Research Park, Norwich NR4 7TJ, UK

*Corresponding author: Department of Genetics, Evolution, and Environment, University College London, Gower Street, London WC1E 6BT, UK; CoMPLEX, University College London, Gower Street, London WC1E 6BT, UK. Email: ucbhpom@ucl.ac.uk

Inversions have been proposed to facilitate local adaptation, by linking together locally coadapted alleles at different loci. Prior work addressing this question theoretically has considered the spread of inversions in "continent-island" scenarios in which there is a unidirectional flow of maladapted migrants into the island population. In this setting, inversions capturing locally adaptive haplotypes are most likely to invade when selection is weak, because stronger local selection (i) more effectively purges maladaptive alleles and (ii) generates linkage disequilibrium between adaptive alleles, thus lessening the advantage of inversions. We show this finding only holds under limited conditions by studying the establishment of inversions in a more general two-deme model, which explicitly considers the dynamics of allele frequencies in both populations linked by bidirectional migration. In this model, the level of symmetry between demes can be varied from complete asymmetry (continent-island) to complete symmetry. For symmetric selection and migration, strong selection increases the allele frequency divergence between demes thereby increasing the frequency of maladaptive alleles in migrants, favoring inversions—the opposite of the pattern seen in the asymmetric continent-island scenario. We also account for the likelihood that a new inversion captures an adaptive haplotype in the first instance. When considering the combined process of capture and invasion in "continent island" and symmetric scenarios, relatively strong selection increases inversion establishment probability. Migration must also be low enough that the inversion is likely to capture an adaptive allele combination, but not so low as to eliminate the inversion's advantage. Overall, our analysis suggests that inversions are likely to harbor larger effect alleles that experience relatively strong selection.

Keywords: inversions; local adaptation; structural variants; gene flow

## Introduction

Chromosomal inversions are a form of structural variant that suppress recombination between loci. Inversions can result in reduced fitness due to the disruption of genes around their breakpoints (Kirkpatrick 2010), or from the capture and accumulation of deleterious alleles due to their lower effective recombination rate (Wasserman 1968; Berdan et al. 2021). Furthermore, inversion heterozygotes may experience reduced fecundity as a result of improper meiosis that results in aneuploid gametes (White 1978). Despite these negative fitness effects, the prevalence of inversions has led to several putative explanations for their continued persistence (see reviews Kirkpatrick 2010; Wellenreuther and Bernatchez 2018; Faria et al. 2019b; Huang and Rieseberg 2020; Villoutreix et al. 2021; Berdan et al. 2023). In particular, theoretical models have demonstrated how inversions could facilitate local adaptation under gene flow by increasing linkage between coadapted alleles and reducing effective migration of maladapted haplotypes (Kirkpatrick and Barton 2006; Charlesworth and Barton 2018).

Empirical evidence for this hypothesis has since been documented across a wide array of taxa (e.g. Lowry and Willis 2010; Cheng et al. 2012; Ayala et al. 2013; Lee et al. 2017; Christmas et al. 2019;

Faria et al. 2019a; Huang et al. 2020; Koch et al. 2021; Hager et al. 2022; Harringmeyer and Hoekstra 2022), and a body of related theoretical work has also developed from the original model, investigating the roles of geography, chromosome type, and inversion length on the fate of adaptive inversions (Feder et al. 2011; Charlesworth and Barton 2018; Connallon et al. 2018; Connallon and Olito 2021; Proulx and Teotónio 2022). For simplicity, this work often considers a "continent-island" model, in which inversions are introduced into an "island" population which receives maladapted migrants from a larger "continent" population. In this model, the selective advantage of an adaptive inversion is proportional to the rate of gene flow (Kirkpatrick and Barton 2006), and inversely proportional to the strength of selection on the island (Bürger and Akerman 2011; Charlesworth and Barton 2018). These results rely on the homogeneous maladaptation of migrant alleles which follows from the extreme migration asymmetry assumed between the continent and island populations (Kirkpatrick and Barton 2006). This scenario is unlikely to apply to many empirical systems, where local adaptation occurs in a structured population with migration between similarly sized populations at similar rates (e.g. Feder et al. 2011; Proulx and Teotónio 2022). Proulx and Teotónio (2022) analyzed the strength of selection on rare modifiers of recombination in a two-patch model with symmetric local selection and

migration, allowing for variation in the number of selected loci and epistasis. They show there is a positive relationship between the strength of selection for local adaptation and selection for a locally adaptive inversion and provide an expression for the inversion's advantage in terms of the genetic load present in the population. However, framing the advantage in terms of load does not allow the separation of the effects of different model parameters and their interactions. In particular, with two-way dispersal, selection will interact with migration to determine the overall rate of maladaptive gene flow. In order to understand what forces drive selection on inversions, it is necessary to look individually at the roles of recombination, migration, selection for local adaptation, and the interactions between them.

In addition to population structure, it is important to consider all steps in the establishment of an inversion. This includes not only whether an existing locally adaptive inversion spreads but also how the frequency of adaptive haplotypes affects their probability of being captured by an inversion in the first place. Assuming an inversion captures a random genotype, the probability it contains a particular adaptive combination is proportional to the frequency of that haplotype. This has been briefly discussed before Kirkpatrick and Barton (2006), and in comparisons of X-linked and autosomal inversions (Connallon *et al.* 2018). But so far models have sidestepped the problem by assuming that either an inversion capturing the coadapted haplotype simply existed (Kirkpatrick and Barton 2006; Charlesworth and Barton 2018; Connallon *et al.* 2018; Connallon and Olito 2021) or that such an inversion arose during a period of allopatry (Feder *et al.* 2011). Explicitly modeling the origin of the inversion is important because regimes favorable for the invasion of an adaptive inversion are not necessarily those where coadapted genotypes are common. For example, previous models show adaptive inversions are expected to be favored most when there are high rates of migrant gene flow (Kirkpatrick and Barton 2006; Charlesworth and Barton 2018). However, in this case there are fewer fit genotypes to be captured, so migration can have opposing effects on the probability of capture and invasion.

Here, we model the fate of locally adaptive chromosomal inversions in a two-locus, two-allele, two-deme model with migration and selection. We consider the case of symmetrical deme sizes and migration, as well as asymmetrical scenarios with the continent-island model as the extreme case. To understand when inversions are favored in this model, we determine their selective advantage in a population in which the locally adaptive alleles have reached their equilibrium frequencies and linkage disequilibrium under migration and selection. By considering the process of inversion capture as well as invasion, we determine population structures which favor the establishment of locally adaptive inversions under environmentally variable selection.

## Methods

We consider a population consisting of two demes linked by bidirectional migration with selection for different locally adapted alleles in each deme. We start by deriving analytical expressions for equilibrium allele frequencies at the local adaptation loci and the linkage disequilibrium (LD) between them. This will allow us to assess the frequency of each haplotype and hence the initial rate of increase of an inversion capturing a locally adapted combination of alleles. We then determine the probability of such an inversion arising in the first place and, combining capture and invasion, analyze the pattern of establishment of adaptive inversions in the population.

## Model

We model a population of two demes, each consisting of an infinite number of haploid, hermaphroditic individuals, with nonoverlapping generations. The model is equally applicable to a case with two sexes at even sex ratio, as long as the genetic determination of sex is unlinked to the adaptive loci under consideration. Selection acts on two loci, $A$ and $B$, that each have two alleles $A_i$ and $B_j$, where $i, j \in \{1, 2\}$. Both loci contribute to a trait under selection for local adaptation; the subscript of an allele denotes the deme in which it provides a benefit $s_i$ or $s_j$ (equal for each of the two loci). Fitness is multiplicative so that the relative fitness of an individual in deme $i$ is either $(1 + s_i)^2$, $(1 + s_i)$ or 1, depending on whether it carries two, one, or no allele(s) conferring local adaptation to its environment.

The life cycle begins with adults. These individuals reproduce, whereby for each offspring a pair of parents is sampled according to their relative fitness in their current deme. During reproduction, recombination occurs at rate $r$. When alleles are held in an inversion, the recombination rate with noninverted chromosomes reduces to zero (double cross-overs and gene conversion are ignored). Migration between demes then occurs such that a proportion $m_{lk}$ of juveniles in deme $k$ are migrants from deme $l$. After migration, the juveniles in each deme become the adults of the next generation. As the life cycle consist of just two phases, reproduction/selection and dispersal, the order of events within a generation does not affect the results.

At the beginning of a generation, adults with genotype $A_iB_j$ in deme $k$ occur at frequency $p_{ij}^k$ and have fitness $w_{ij}^k$. Among the parents sampled for reproduction, the frequencies are $\tilde{p}_{ij}^k = p_{ij}^k(w_{ij}^k/\bar{w}_k)$, where $\bar{w}_k$ is the mean fitness in deme $k$. Similarly, $D_k = p_{11}^k p_{22}^k - p_{12}^k p_{21}^k$ is the coefficient of linkage disequilibrium among all adults in deme $k$, and $\tilde{D}_k = \tilde{p}_{11}^k \tilde{p}_{22}^k - \tilde{p}_{12}^k \tilde{p}_{21}^k$ is the linkage disequilibrium among parents, i.e. after selection. Among the juveniles of the next generation, the frequency of genotype $A_iB_j$ in deme $k$ after migration, is given by

$$p_{ij}^{k'} = (1 - m_{lk})(\tilde{p}_{ij}^k - r\tilde{D}_k) + m_{lk}(\tilde{p}_{ij}^l - r\tilde{D}_l) \tag{1}$$

if $i = j$, and

$$p_{ij}^{k'} = (1 - m_{lk})(\tilde{p}_{ij}^k + r\tilde{D}_k) + m_{lk}(\tilde{p}_{ij}^l + r\tilde{D}_l) \tag{2}$$

otherwise.

For analytical tractability, we convert this discrete time system to continuous time by assuming all rate parameters ($m$, $s$, $r$) are small and of the same order. When migration is limited to one direction (i.e. $m_{12}$ or $m_{21} = 0$) or when selection in one environment is very strong ($s_i \gg s_j$), the model approaches the well studied "continent-island" model (hereafter superscript "C-I," e.g. Kirkpatrick and Barton 2006; Charlesworth and Barton 2018).

## Analysis

To calculate the equilibrium allele frequencies before the spread of an inversion, we use a quasi-linkage equilibrium (QLE) approximation. This assumes that recombination between the two loci is sufficiently high compared to migration and selection so that LD reaches an equilibrium more quickly than the allele frequencies. This approximation is appropriate because we are interested in the spread of inversions that suppress recombination so we can disregard cases where the recombination rate is already small

and inversions have little effect. To ensure the existence of a migration-selection balance equilibrium, we assume migration is weak compared to selection (i.e. $\max(m_{12}, m_{21}) < \min(s_1, s_2)$). Once we calculate the equilibrium genotype frequencies in each deme, we can derive the rate of increase of a rare adaptive inversion. We describe the method assuming that the inversion captures the $A_1B_1$ haplotype—the method for inversions capturing alternative haplotypes is analogous.

We first rewrite the genotype frequencies in terms of allele frequencies and LD and calculate their equilibria (Otto and Day 2011). So, define $f^k_{A_1}$ and $f^k_{B_1}$ respectively as the frequencies of alleles $A_1$ and $B_1$ in deme $k$. The other allele frequencies are given by $f^k_{A_2} = 1 - f^k_{A_1}$. Using these and Equations 1 and 2 with $r = 0$, the dynamics of an $A_1B_1$ inversion are then described by the matrix $C_{A_1B_1}$, in which the entry $c_{kl}$ is the expected number of offspring an inverted parent in deme $k$ has that are in deme $l$ at the end of the generation.

$$C_{A_1B_1} = \begin{pmatrix} \frac{(1-m_{21})(1+s_1)^2}{\hat{w}_1} & \frac{m_{12}(1+s_1)^2}{\hat{w}_1} \\ \frac{m_{21}}{\hat{w}_2} & \frac{(1-m_{12})}{\hat{w}_2} \end{pmatrix}, \tag{3}$$

where $\hat{w}_k$ is the equilibrium mean fitness in deme $k$, calculated from the allele frequencies at QLE (we denote equilibrium values with the hat symbol ^ throughout). The rate at which a rare $A_1B_1$ inversion increases in frequency in the whole population is given by the leading eigenvalue of $C_{A_1B_1}$ ($\lambda_{A_1B_1}$). As the population is at equilibrium, the growth rate of a recombining $A_1B_1$ haplotype is 1, so $\lambda_{A_1B_1} > 1$ implies a benefit to the inversion that can be ascribed to the absence of recombination, and $\lambda_{A_1B_1} - 1$ is the initial selection coefficient of the inversion.

In continent-island models it is possible to deduce the invasion probability of an inversion from the initial rate of increase alone. This is because inversions cannot migrate from the island to the continent and selection on rare inversions is therefore approximately constant. So, one can use asymptotic approximations of invasion probabilities based on this rate (Otto and Whitlock 2013). In the two-deme model, inversions can originate in either deme and migrate between demes. While the inversion is rare, this migration can induce early extinction events that would not happen in the continent-island scenario. As a result the inversion is less likely to reach the rate of asymptotic growth, given by $\lambda_{A_1B_1} - 1$. Using the eigenvalue would therefore overestimate the probability of invasion. This effect is greatest when rates of migration are relatively large (see Supplementary Fig. S1). We account for this by modeling the inversion's spread with a two-type branching process. To this end, define $u_i$ as the probability of invasion conditional on the inversion arising in deme $i$. The probabilities $u_i$ can be derived by analyzing the branching process corresponding to the mean matrix $C_{A_1B_1}$ (see Supplementary Information). Then, the probability of invasion given that the inversion captures $A_1B_1$ is $d_1u_1 + d_2u_2$, where $d_i$ is the proportion of $A_1B_1$ haplotypes that are in deme $i$. Regardless, it is still useful to derive $\lambda_{A_1B_1}$ because forces governing this overall initial rate of increase are likely to be similar to those determining the invasion probability.

### Capture of locally adaptive alleles

Above, we analyzed the dynamics of locally adaptive inversions. In order to be locally adaptive, however, an inversion must have captured a locally adaptive haplotype—the chance of this occurring depends on the frequency of said haplotype in each

population. Now, the probability that an inversion captures coadapted alleles ($A_iB_i$) and invades is given by

$$\gamma_{ii} = u_1 p^1_{ii} + u_2 p^2_{ii}, \tag{4}$$

because $p^k_{ii}$ is the probability the inversion arises in deme $k$ and captures $A_iB_i$. Finally, the probability of any locally adaptive inversion establishing when it arises needs to consider both $A_1B_1$ and $A_2B_2$ haplotypes, and is given by

$$\Gamma = \gamma_{11} + \gamma_{22}. \tag{5}$$

This is also equal to the probability of an inversion establishing itself overall, because inversions that capture allele combinations that are not advantageous in either deme (i.e. $A_1B_2$ or $A_2B_1$) are never favored.

## Results

### Equilibrium allele frequencies and linkage disequilibrium

We proceed by first calculating the allele frequencies at linkage equilibrium, when $r$ is large, and then perturbing the small term $1/r$ to obtain expressions at quasi-linkage equilibrium (Akerman and Bürger 2014). First, we define the ratio of migration to selection as $\alpha_i = m_{ij}/s_j < 1$ and

$$\begin{aligned} \hat{f}^1_0 &= \tfrac{1}{2}\left(1 - 2\alpha_1 + \sqrt{1 + 4\alpha_1\alpha_2}\right), \\ \hat{f}^2_0 &= \tfrac{1}{2}\left(1 + 2\alpha_2 - \sqrt{1 + 4\alpha_1\alpha_2}\right), \end{aligned} \tag{6}$$

which respectively are the frequencies of allele $A_1$ in demes 1 and 2 at linkage equilibrium ($D = 0$). At the quasi-linkage equilibrium, the frequencies of the alleles ($\hat{f}^i_j$ for allele $j$ in deme $i$) are

$$\begin{aligned} \hat{f}^1_{A_1} &= \hat{f}^1_{B_1} \approx \hat{f}^1_0 + \frac{\alpha_1}{r}\left(f^1_0 - f^2_0\right)\left(s_1 - \frac{\alpha_2(s_1+s_2)}{\sqrt{1+4\alpha_1\alpha_2}}\right) + O(r^{-2}), \\ \hat{f}^2_{A_1} &= \hat{f}^2_{B_1} \approx \hat{f}^2_0 - \frac{\alpha_2}{r}\left(f^1_0 - f^2_0\right)\left(s_2 - \frac{\alpha_1(s_1+s_2)}{\sqrt{1+4\alpha_1\alpha_2}}\right) + O(r^{-2}), \end{aligned} \tag{7}$$

and the linkage disequilibrium between loci in deme 1 ($D_1$) is

$$\begin{aligned} \hat{D}_1 &\approx \frac{m_{21}\left(\hat{f}^1_0 - \hat{f}^2_0\right)^2}{r} \\ &\approx \frac{m_{21}}{r}\left(\alpha_1 + \alpha_2 - \sqrt{1 + 4\alpha_1\alpha_2}\right)^2 + O(r^{-2}). \end{aligned} \tag{8}$$

Linkage disequilibrium in deme 2 ($\hat{D}_2$) is given by replacing $m_{21}$ with $m_{12}$ and vice versa. These equilibrium values, derived here for haploidy, are in accord with previous results (Akerman and Bürger 2014).

In the case where migration and selection are symmetric, $m_{kl} = m$ and $s_i = s$ (i.e. two populations with exactly opposing local selection pressures exchanging an equal proportion of migrants), the demes have symmetric allele frequencies ($f^2_{A_1} = \hat{f}^2_{B_1} = 1 - \hat{f}^1_{A_1} = 1 - \hat{f}^1_{B_1}$) and linkage disequilibria ($D_1 = D_2$)

$$\hat{f}^1_{A_1} = \hat{f}^1_{B_1} \approx \frac{1}{2}\left(1 - 2\alpha + \sqrt{1 + 4\alpha^2} - \frac{m}{r}\left(8\alpha - 2\frac{1 + 8\alpha^2}{\sqrt{1 + 4\alpha^2}}\right)\right) + O(r^{-2}), \tag{9}$$

$$\hat{D} \approx \frac{m}{r}\left(\sqrt{1 + 4\alpha^2} - 2\alpha\right)^2 + O(r^{-2}). \tag{10}$$

In the other extreme case, where there is unidirectional gene flow from deme 2 ("continent") to deme 1 ("island"), the "continent" genotypes remain fixed to $A_2B_2$. Setting $s_1 = s$ and $m_{21} = m$

$$\hat{f}_{A_1} = \hat{f}_{B_1} \approx \left(1 - \frac{m}{s}\right)\left(1 + \frac{m}{r}\right) + O(r^{-2}), \quad (11)$$

$$\hat{D} \approx \frac{m}{r}\left(1 - \frac{m}{s}\right)^2 + O(r^{-2}), \quad (12)$$

as in Bürger and Akerman (2011). Locally adaptive alleles are more abundant in the symmetric scenario (equation 9) than in the continent-island scenario (equation 11). This difference arises because in the symmetric scenario a fraction of locally adapted migrants from a focal deme migrate to and survive in the other deme, only for their offspring to return back and contribute to the frequency of beneficial alleles in the focal deme. In the continent-island scenario, in contrast, continental migrants can only introduce maladaptive alleles into the focal deme.

In both scenarios, linkage disequilibrium is positive, indicating that the adaptive alleles tend to be found together in coadapted haplotypes ($A_1B_1$ and $A_2B_2$). This tendency increases with the strength of selection in both models ($\partial\hat{D}/\partial s \geq 0$), because selection generates an association between coadapted alleles but decreases with the rate of recombination ($\partial\hat{D}/\partial r \leq 0$) which breaks the coadapted haplotypes apart to create intermediate haplotypes ($A_1B_2$ and $A_2B_1$).

The role of migration is less straightforward and differs between the two scenarios. At low migration rates, selection tends to be stronger relative to migration and demes are enriched for locally adapted haplotypes. Linkage disequilibrium then increases with $m$ because more $A_2B_2$ combinations are introduced into deme 1 (and more $A_1B_1$ combinations are introduced into deme 2 in the symmetric scenario), which allows the creation of the recombinant genotypes $A_1B_2$ and $A_2B_1$. When migration becomes higher, it degrades the linkage disequilibrium that is built up locally by selection by reducing the divergence in allele frequencies between demes (equation 8). In the extreme, high rates of migration homogenize the population by overriding population structure. The rate of migration at which this effect sets in depends on the model. In the continent-island scenario, migration decreases linkage disequilibrium when $m > s/3$. In the symmetric case, migration begins to decrease linkage disequilibrium at a lower point, when $m > s\sqrt{3}/6$, because the presence of $A_1B_1$ migrants in deme 2 generates more intermediate haplotypes through recombination. These individuals can back-migrate and degrade linkage disequilibrium in deme 1 (with the same process going on in the reverse direction).

## Invasion probability of a locally adaptive inversion

Having established the equilibrium composition of populations, we can now consider the fate of a new inversion that captures alleles $A_1$ and $B_1$, which are locally adaptive in deme 1. We first calculate the initial rate of increase of the inversion, comparing the continent-island and the symmetric scenarios before the full model. In the first two cases, we derive expressions for the selective advantage of the inversion assuming quasi-linkage equilibrium—the full model proved too complex to yield analytical predictions. Comparisons between the approximations and the exact numerical solution can be found in the Supplementary Mathematica file (File S1) and pdf (File S2). The invasion probability cannot be extrapolated from the initial selective advantage in the two-deme case, so we use a multitype branching process (see Supplementary Information). Analytical solutions were unobtainable from this method, so for a fair comparison we contrast numerical solutions for both cases.

Under QLE assumptions and with $s > m$, the initial rate of increase of the inversion in the continent-island scenario is

$$s_{inv}^{C-I} \approx \frac{mr}{r + 2s - 2m}, \quad (13)$$

as given by equation A11 in Charlesworth and Barton (2018), which decreases with $s$ but increases with $m$ and $r$. Under the sole assumption that migration is weak, the advantage is given by $mr/(r + 2s)$, as in Bürger and Akerman (2011). The selective advantage of the inversions stems from its ability to maintain linkage between alleles $A_1$ and $B_1$. Stronger selection reduces the advantage of an inversion because selection already generates positive linkage disequilibrium between the two locally adaptive alleles. Accordingly, when recombination rates are high, the inversion maintains most of its advantage even when selection is strong—any LD built by selection is rapidly eroded by recombination of noninverted chromosomes.

In a two-deme model, stronger selection again removes maladaptive alleles from the focal deme but also increases their frequency among migrants. The initial selective advantage of the inversion in the symmetric case is

$$s_{inv}^{symm} \approx m - s\left(A - \sqrt{1 + \alpha^2}\right) - \frac{m}{r}\left(\frac{s(4A^2 - 2)}{A} - 8m\right) + O(r^{-2}), \quad (14)$$

where $A = \sqrt{1 + 4\alpha^2}$ ($\alpha = m/s < 1$). The first two terms of equation 14 are independent of $r$ and give the inversion's rate of increase when the population is at linkage equilibrium. This increases with $s$ (which drives the bracketed second term to zero) and tends towards $m$, reflecting the effect of stronger local selection on the composition of migrant gene flow. The third term (with factor $1/r$) relates to the effect of LD within the population. As strong selection builds LD, this lessens the advantage of the inversion (the bracketed term involving $A$ is always positive). In contrast, high rates of recombination break down LD, weakening the effect of selection on the buildup of LD, giving greater advantage to the inversion. The relative contribution of selection thus depends on the recombination rate. Equation 14 can also be expressed as $L - s((1 + \alpha) - \sqrt{1 + \alpha^2}) \approx L - m$ when $m \ll s$, where $L(m, s, r) = 1 - \bar{w}/(1 + s)^2$ is the genetic load present in one of the demes, as in Proulx and Phillips (2005) and Proulx and Teotónio (2022).

The contrasting effects of selection can be seen from the numerically solved invasion probabilities (Fig. 1). Contrary to the continent-island model, where selection on the inversion decreases with the strength of selection, stronger selection can favor the invasion of locally adaptive inversions in the symmetric scenario. This happens because selection reinforces local adaptation so migrants bring more maladaptive alleles into the focal deme, which can increase the rate at which recombination degenerates LD between coadapted alleles. This advantage eventually decreases as the strength of selection increases, because this positive effect becomes outweighed by selection both eliminating migrant genotypes and maintaining LD between locally adaptive alleles, reducing the impact of recombination suppression. When selection is very strong relative to migration, the invasion probability under symmetric migration converges to that of the island-continent scenario (Fig. 1), because migrants tend to carry exclusively maladaptive alleles as in the continent-island scenario. The two-deme model also allows us to include asymmetric local selection and migration (Fig. 2) and thus explore intermediates between the symmetric and the continent-island models. Selection in the focal deme ($s_1$) increases the degree of local

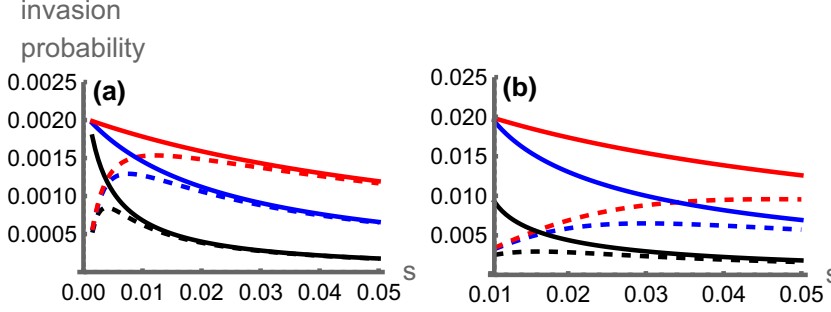

**Fig. 1.** Numerically calculated invasion probabilities for an inversion capturing $A_1B_1$ against the strength of selection ($s$) in favour of this haplotype, for the symmetric (dashed lines) and continent-island (solid lines) scenarios with $r = 0.01$ (black), $r = 0.05$ (blue), or $r = 0.15$ (red), which have lower to higher invasion probabilities. In panel A, $m = 0.001$; in panel B $m = 0.01$. Data with $s < m$ are excluded as the adaptive alleles may not be at a stable equilibrium.

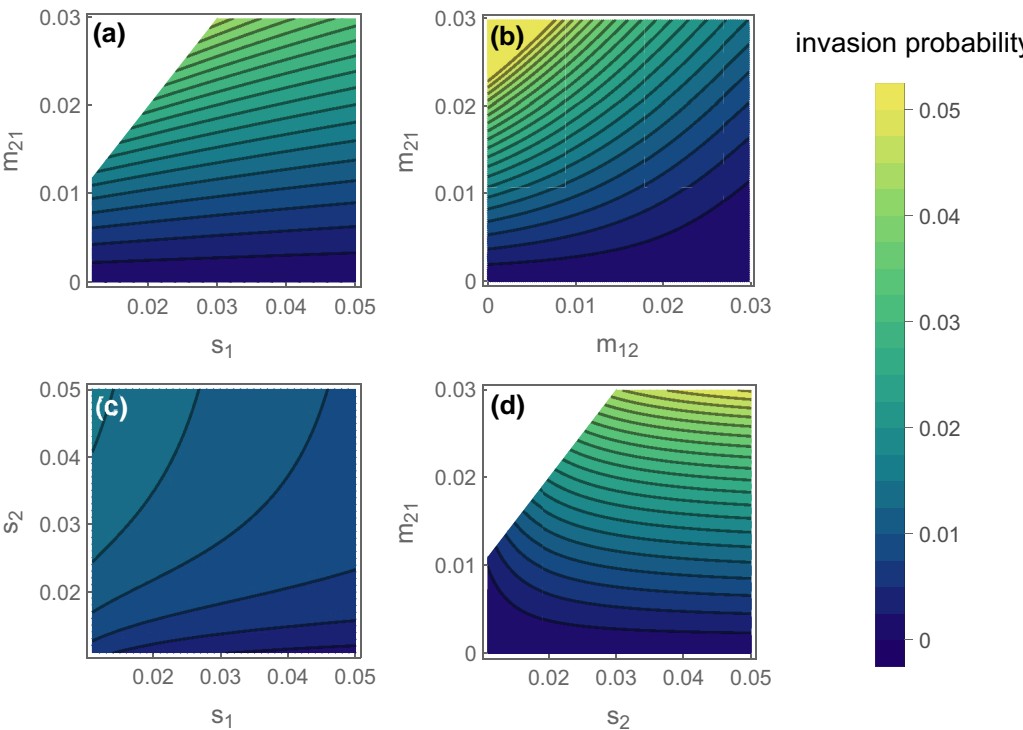

**Fig. 2.** Numerically calculated invasion probability of an inversion that arises on an $A_1B_1$ individual. In each panel, migration parameters that are not varying along an axis had a default value of 0.01, and selection parameters had a default value of 0.03. Recombination was set to r = 0.1.

adaptation and inversions therefore have a lesser advantage. This effect is strongest when there are more maladapted migrants entering deme 1 (higher $m_{21}$, Fig. 2a) or when the genotypic composition of migrants is more maladapted (higher $s_2$, Fig. 2c) but has a weaker effect on inversion invasion probability than parameters that change the genotypic composition of migrants ($m_{21}$ and $s_2$). For a fixed level of migration into deme 1 ($m_{21}$), invasion probability decreases with increasing migration out of deme 1 ($m_{12}$) because inversions migrate out of the environment in which they are adapted (Fig. 2b). Overall, increased migration from and selection in deme 2 are the most important factors in generating the inversion's advantage (Fig. 2d)—exactly the two parameters that are most extreme in the continent-island model.

### Combined capture and invasion probability of locally adaptive inversions

The analysis above calculates the invasion probability assuming that an inversion has captured the $A_1B_1$ haplotype. It does not take into account the probability of this actually happening, i.e. of an inversion occurring in an $A_1B_1$ individual to capture both locally adapted alleles together. We term the combined process of the capture and subsequent invasion as the "establishment." As the strength of selection s increases, more locally adaptive $A_1B_1$ genotypes are available to be captured by an inversion (Fig. 3). The resulting increased capture rate has a larger positive effect on the establishment probability than the negative impact of more intense selection on the inversion's subsequent selective advantage relative to the population (as illustrated in Fig. 1). Thus, our results predict that stronger selection is more likely to drive the evolution of locally adaptive inversions. Importantly, this is true for both scenarios and qualitatively alters the prediction for how inversions should contribute to local adaptation in the continent-island scenario (c.f. Fig. 1).

We can also see how asymmetric migration or selection affect the establishment probability. While high migration into deme 1 strongly favors the invasion of existing adaptive inversions (Fig. 2b), the lower frequency of coadapted haplotypes also lowers

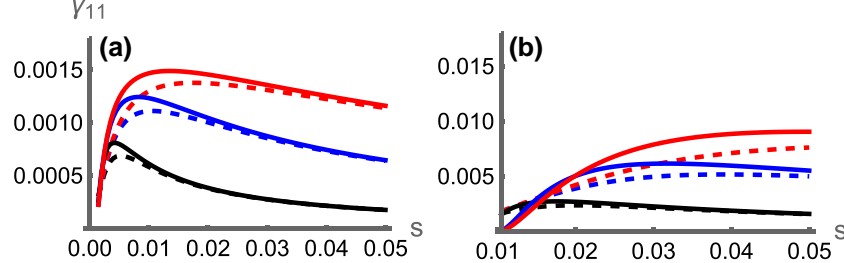

**Fig. 3.** Combined probability of an inversion arising on an $A_1B_1$ haplotype and then invading ($\gamma_{11}$) against the strength of selection (s), for the symmetric (dashed lines) and continent-island (solid lines) scenarios with $r = 0.01$ (black), $r = 0.05$ (blue), or $r = 0.15$ (red). In (A), $m = 0.001$; in (B), $m = 0.01$. Data with $s < m$ are excluded as the adaptive alleles may not be at a stable equilibrium.

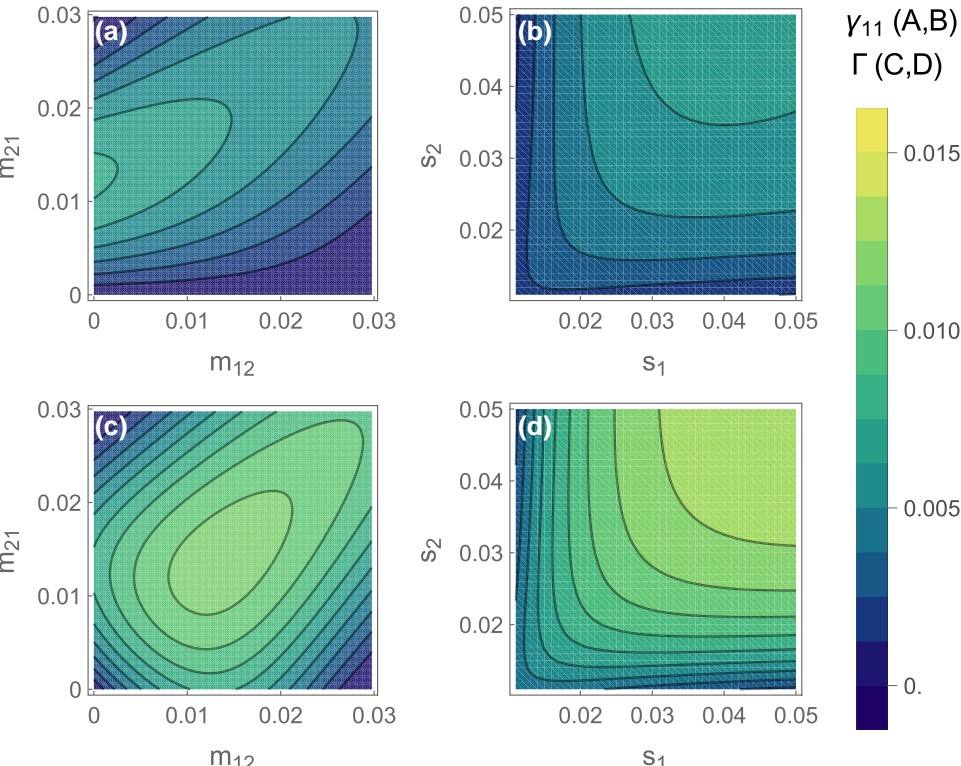

**Fig. 4.** Total establishment probability of an adaptive inversion across the whole population. (A, B) Combined probability of an inversion arising on the $A_1B_1$ haplotype and then invading ($\gamma_1$) for asymmetric migration (A) or selection (B). (C, D) Probability of an inversion capturing either adaptive haplotype ($A_1B_1$ or $A_2B_2$) and invading ($\Gamma$) for asymmetric migration (C) or selection (D). The continent-island model corresponds to $m_{12} = 0$ (Y axis in A, C) and the symmetric two-deme model corresponds to the $s_1 = s_2$ diagonal in (B) and (D). In each panel, migration parameters that are not varying along an axis had a default value of 0.01, and selection parameters had a default value of 0.03. Recombination was set to $r = 0.1$. To ensure stability, we vary parameters in the range where $\max(m_{12}, m_{21}) < \min(s_1, s_2)$, $r = 0.1$.

the probability of them arising in the first place. Thus, adaptive inversions are most likely to form and invade when $m_{21}$ is intermediate, such that the probability of an inversion capturing an adaptive haplotype and the inversion's subsequent selective advantage are both reasonably large (Fig. 4a).

Increasing the strength of selection in either deme typically increases the chance that adaptive inversions will arise and spread. Increasing the strength of selection in deme 2 ($s_2$) increases migration load and therefore the inversion's advantage and increasing selection in deme 1 ($s_1$) increases the probability of capturing the adaptive haplotype (Fig. 4b). Yet, as discussed above, $A_1B_1$ inversion invasion probabilities decline under very strong selection in deme 1 (very high $s_1$) by increasing pre-existing adaptation. Nevertheless, stronger local selection usually creates a more favorable environment for establishment.

So far, we have only considered the evolution of a specific inversion, adaptive in one deme. This is the only plausible scenario in the continent-island scenario, where only inversions that capture the island-adapted haplotype $A_1B_1$ are of interest. However, with two demes, divergent local adaptation can occur from either adaptive inversion, both due to the beneficial effects in the favored deme and due to the protection from recombination that such an inversion offers to individuals adapted to the other deme. So in this final section we consider the overall probability of local adaptation through the spread of an inversion that arises anywhere in the population ($\Gamma = \gamma_{11} + \gamma_{22}$; Fig. 4c, d).

Under symmetric local selection, inversions are most likely to establish when migration is symmetric and intermediate (Fig. 4c). Migration rates that are favorable for the establishment of inversion in one deme are not so favorable in the other

($\gamma_2$ values can be seen by reflecting Fig. 4a, b across the diagonal) such that symmetric migration rates give the highest overall probability of inversion establishment. Similarly, when migration is symmetric, strong and symmetric local selection is most conducive to the formation and spread of locally adaptive inversions (Fig. 4d). Across both demes, this maximizes the probability of capturing an adaptive haplotype while maintaining the migration load that is necessary to drive invasion.

## Discussion

Here, we have examined the evolution of locally adaptive chromosomal inversions while explicitly modeling selection across a structured population. Inversions can keep locally favored allele combinations together in the face of potential recombination with maladapted migrants. Therefore, locally adaptive inversions spread fastest when migrant alleles are homogeneously maladaptive, as assumed in the continent-island scenario that has been well studied (Kirkpatrick and Barton 2006; Charlesworth and Barton 2018). The continent-island scenario represents an extreme, where migrants are fixed in their genetic composition for being purely maladaptive. In comparison, the two-deme model leads to a number of novel insights. First, the interplay of selection and migration in both populations can mean that stronger local selection favors inversions, unlike in the continent-island scenario (Fig. 1). Second, we can examine asymmetric selection pressures across demes, showing that increasing the strength of selection in the deme with the weakest selection for local adaptation generally promotes the establishment of adaptive inversions, by either increasing the selective advantage or the probability of capture (Fig. 4). Thus, symmetric patterns of migration and selection are the most conducive to inversion establishment. By considering the probability that inversions initially capture favorable haplotypes, we show that relatively strong selection is most likely to underlie inversions in any of the population structures (Fig. 3). Overall, our results show that inversions are particularly likely to establish when selection on locally adaptive alleles is strong, contrary to what previous analyses would suggest.

Migration regimes under which inversions are likely to form and spread are fairly specific because they must satisfy multiple requirements. Firstly, we assume that locally adaptive alleles are polymorphic, which means they must be able to resist swamping by migration. This condition requires relatively weak migration and is likely to be a significant constraint on the existence of local adaptation (Feder et al. 2011). Then, given that locally adaptive alleles are maintained, higher migration rates favor the spread of inversions because they increase the frequency of the maladaptive alleles and thus the cost of recombination (Figs. 1, 2). However, this also has the effect of reducing the frequency of adaptive haplotypes so that inversions are less likely to capture pure sets of adaptive alleles (Fig. 4). The result is that higher migration rates do not always favor the evolution of inversions. In general, rates of migration may turn out to restrict the evolution of locally adaptive inversions more than previously thought.

Inversions are favored because they reduce recombination so they have the strongest advantage when the initial recombination rate is high. The positive relationship between the strength of selection and invasion or establishment probability is also most robust when the rate of recombination between adaptive loci is not too small ($r > 0.05$). Such values are not unrealistic—for example, in *Littorina saxatilis* putative inversions associated with differentiation between ecotypes were 12.5–29.5 cM (corresponding to $r \approx 0.11 - 0.22$ between the endpoints) (Westram et al. 2018; Faria

et al. 2019a), and similar putative inversions in *Littorina fabalis* span genetic distances of 0.6–47 cM ($r \approx 0.006 - 0.3$) (Le Moan et al. 2024). These values provide a conservative upper bound for $r$, since any locally adaptive alleles are unlikely to both be close to the inversion breakpoints.

Theories concerning the origins of adaptive inversions can broadly be split into three categories (Schaal et al. 2022): "capture," in which an inversion creates a linkage group of existing adaptive variation and spreads (Kirkpatrick and Barton 2006); "gain," in which an inversion is initially polymorphic (e.g. due to drift, underdominance, or acquisition of a good genetic background) and then accumulates adaptive variation which is subsequently protected from recombination (e.g. Lamichhaney et al. 2016; Samuk et al. 2017); or "generation," in which adaptive variation is created when the inversion arises through the breakpoint creating beneficial new coding sequence or gene expression variants (Feder and Nosil 2009; Villoutreix et al. 2021, e.g. Jones et al. 2012). Our work focuses on the "capture" hypothesis in which locally adaptive alleles are already segregating and have reached migration–selection equilibrium and may have already evolved enhanced local fitness. There is *a priori* no reason why any inversion with "capture" origins could not subsequently gain more adaptive variation at a later date as set-out in the "gain" hypothesis. But we show that already in a pure "capture" scenario, large effect alleles are the most likely to underlie adaptive inversions, as large selection coefficients are more favorable for the establishment of inversions.

Indeed, theory suggests the effect size distribution of locally adaptive alleles is likely to be skewed towards those that are strongly selected, which could thus facilitate the evolution of adaptive inversions. In the short term, locally adaptive alleles must experience fairly strong selection to be able to resist being swamped by migration (Lenormand 2002; Yeaman 2015). Small effect alleles can still contribute to local adaptation when they arise in close linkage with large effect alleles, resulting in aggregated regions of adaptation which could be modeled as a single locus of large effect (Yeaman and Whitlock 2011). Alternatively, they can contribute transiently before being lost (Yeaman 2015). With high gene flow, and over long timescales, the architecture of local adaptation is expected to evolve towards a few, highly concentrated clusters of small effect alleles linked with large effect alleles (Yeaman and Whitlock 2011), which are likely to be particularly conducive to inversion establishment.

Schaal et al. (2022) used simulations to study the invasion of inversions capturing variation that influences a polygenic quantitative trait, finding that inversions involved in local adaptation tended to exhibit more of a capture than a gain effect when alleles were unlikely to be swamped. When alleles were prone to swamping by migration, persisting locally adaptive inversions had often gained much more adaptive variation post-capture. Under high rates of gene flow both capture and gain scenarios are plausible, depending on the effect size of the loci captured. Because adaptive alleles can be gained after the inversion arose and spread, recent inversions may offer the best opportunity to test our predictions about the effect size of alleles driving the evolution of locally adaptive inversions. The allelic content of such inversions could depend on how long the populations in question have been diverging, with the expectation that long periods of divergence results in a more concentrated architecture (Yeaman and Whitlock 2011). However, separating the individual trait effects of different loci within the inversion is challenging once they have been linked together. Thus, despite the prevalence of putatively adaptive inversions, mapping of quantitative trait loci has been achieved in only a handful of cases (e.g. Peichel and Marques 2017; Koch et al. 2021;

Poelstra *et al.* 2014 for an example unrelated to local adaptation) leaving open questions about the number and effect size of loci that underpin inversion selective advantage (Tigano and Friesen 2016).

We only consider the evolution of inversions that link alleles at two relatively nearby loci. It is possible that more loci contribute to local adaptation and an inversion could capture more than two loci. When loci are more numerous and of individually weaker effect, such that their total combined effect size is the same, populations are on average less well adapted (Proulx and Teotónio 2022). Inversions capturing a full complement of adaptive alleles therefore have a relatively greater advantage. However, at the same time it also becomes less likely that an inversion will capture all the adaptive alleles on the same haplotype, since this is rare. Inversions will still spread if they capture more locally adaptive alleles than the population mean, though in this case, the establishment of the inversion could lead to the loss of those adaptive alleles not captured if this arrangement fixes. So, determining inversion establishment probabilities becomes complex when more loci are involved because (i) there are multiple haplotypes that can be favored when inverted, each with different selective advantages, and (ii) the number of genotypes to consider rises exponentially. The relationship between invasion fitness and haplotype frequencies as the number of loci increases remains to be explored, but we expect inversion evolution will continue to depend on a balance between the selective advantage of the captured haplotype and on the probability of capturing a favorable haplotype.

Our model does not include deleterious mutations or breakpoint effects, which can affect the fate of inversions. Low rates of gene flux within inverted arrangements means that deleterious variation captured by the inversion persists for a long time throughout lineages, as purging this variation relies on rare events such as gene conversion and double crossover events. Inversion breakpoints can also disrupt gene function and result in lower individual fitness (White 1978; Kirkpatrick 2010), though this can occasionally be adaptive (e.g. Corbett-Detig 2016). These effects can be incorporated into the model by introducing a fixed cost or benefit. Reduced recombination within inversions severely weakens the efficacy of purifying selection on new mutations (Charlesworth 1996; Betancourt *et al.* 2009). Mutation accumulation is particularly important while the inversion is at low frequency, because most inverted chromosomes will occur in heterokaryotypes where recombination is suppressed (Navarro *et al.* 2000), though gene conversion and double crossover events may alleviate this a little (Berdan *et al.* 2021). In our model, higher recombination rates could correspond to loci located further apart, implying a bigger inversion. For the inversion polymorphism to be maintained, the harmful effect of the mutation load captured by the inversion should be weaker than the advantage provided by the inversion (Nei *et al.* 1967; Connallon and Olito 2021; Berdan *et al.* 2022). However, large inversions will tend to capture more deleterious variation, which may negate the extra benefit the inversion enjoys when recombination rates are high. We model a haploid population, but in diploids the presence and accumulation of strong recessive mutations within inversion will result in negative frequency-dependent selection which limits inversion frequency and the recombination rate (Nei *et al.* 1967; Wasserman 1968; Ohta 1971). The generally deleterious effects associated with inversions likely mean that their invasion probabilities are much lower than we obtain here.

In summary, our results emphasize the likelihood that strongly selected loci can contribute to local adaptation in two ways: by increasing the frequency of adaptive haplotypes that can be captured by an inversion, and by increasing the rate of migrant gene flow and thus the potential cost of recombination. High migration rates also increase this recombination load and thus the selective advantage of an inversion, but this also reduces the frequency of adaptive haplotypes. The probability of adaptive inversion capture could be as important as its selective advantage in determining where such inversions are likely to be found.

## Data availability

A Mathematica notebook containing derivations and code used to generate figures is available as Supplementary File S1 and in a pdf version as Supplementary File S2.

Supplemental material available at GENETICS online.

## Acknowledgments

We are grateful for the helpful feedback provided by two anonymous reviewers, as well as conversations with Bhavin Khatri, Alan Le Moan, and Denis Roze.

## Funding

CM was previously supported by funding from CoMPLEX and an Engineering and Physical Sciences Research Council studentship (EP/N509577/1), and now by a grant from the Gordon and Betty Moore Foundation (GBMF11489). MFS is supported by a Leverhulme Trust Early Career Fellowship (ECF-2020-095). AP is supported by funding from the Engineering and Physical Sciences Research Council (EP/F500351/1, EP/I017909/1), Natural Environment Research Council (NE/R010579/1, NE/X009734/1) and Biotechnology and Biological Sciences Research Council (BB/V003542/1). MR is supported by grants from the Biotechnology and Biological Sciences Research Council (BB/W007703/1, BB/S003681/1 and BB/T019921/1) and the Leverhulme Trust (RPG-2021-414).

## Conflicts of interest

The authors declare no conflict of interest.

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

*Editor: N. Barton*