## [Peer Review File · Genetics]

The establishment of locally adaptive inversions in structured populations

Carl Mackintosh, Michael Scott, Max Reuter, and Andrew Pomiankowski

NOTE: The reviews and decision letters are unedited and appear as submitted by the reviewers.

In extremely rare instances and as determined by a Senior Editor or the EIC, portions of a review may be redacted. If a review is signed, the reviewer has agreed to no longer remain anonymous.

The review history appears in chronological order.

Review Timeline:

Submission Date:	2022-12-05
Editorial Decision:	2023-01-27
Resubmission Received:	2024-01-27
Editorial Decision:	2024-02-27
Resubmission Received:	2024-04-05
Accepted:	2024-04-16

January 26, 2023

GENETICS-2022-305778

The establishment of locally adaptive inversions in structured populations

Dear Dr. Pomiankowski:

Two experts in the field have reviewed your manuscript, and I have read it as well. While both reviewers (and myself) find that the general topic is of interest, reviewer 2 points to a number of important methodological issues (see reviewers comments below). In particular, some of the indices in equations 1-3 seem wrong (this may correspond to typos, but should be checked), and more importantly, the reasoning deriving invasion probabilities from "invasion fitness" (asymptotic rate of increase in the deterministic limit) is unsound, as invasion probability should mostly depend on events occurring during the first generations after the inversion has appeared in a given deme, before the asymptotic rate of increase has been reached. Along the same vein, the reasoning leading to eq. 4 is not justified and does not seem correct. As proposed by the reviewer, a multitype branching process model would probably be the most straightforward way of deriving invasion probabilities in this model. Furthermore, most of the results seem to be obtained from numerical analyses of 2d order approximations, and it is not clear why the authors do not simply perform numerical analyses of the exact model instead. Reviewer 2 also raises doubts on the validity of these 2d order approximations and of the QLE approximations presented in the paper.

While your manuscript is not currently acceptable for publication in GENETICS, we would welcome a substantially revised manuscript addressing the comments of both reviewers. We realize that addressing the methodological issues would imply of substantial amount of extra work, but this would also make the conclusions of the paper much stronger.

We look forward to receiving your revised manuscript. Please let the editorial office know approximately how long you expect to need for revisions.

Upon resubmission, please include:

1. A clean version of your manuscript;
2. A marked version of your manuscript in which you highlight significant revisions carried out in response to the major points raised by the editor/reviewers (track changes is acceptable if preferred);
3. A detailed response to the editor's/reviewers' feedback and to the concerns listed above. Please reference line numbers in this response to aid the editor and reviewers.

Your paper will likely be sent back out for review.

Additionally, please ensure that your resubmission is formatted for GENETICS
<https://academic.oup.com/genetics/pages/general-instructions>

Follow this link to submit the revised manuscript: Link Not Available

Sincerely,

Denis Roze
Associate Editor
GENETICS

Approved by:
Nicholas Barton
Senior Editor
GENETICS

Reviewer #1 (Comments for the Authors (Required)):

The paper uses a well-studied model to show an intuitive but important clarification: that the benefit of an inversion establishing increases with the strength of selection on the locally adapted alleles. Lack of dependence on s is a weird quirk of previous analyses of continent-island model (and their assumptions to solve it) and it is conceptually important to show that other approaches to analysing such models arrive at different conclusions. Another paper has derived analytical machinery that could be used to lead to many of the same findings here (i.e. Akerman and Burger 2014) but it did not actually focus on inversions or

evolution of recombination so the deployment of the model here is still novel and useful. Simulation studies have shown similar findings to those of this paper, but cannot be used to give rule-of-thumb predictions in the same way as analytical models. Given the prevalence of empirical studies on this topic and the murkiness of some of the previous theory papers, this paper is well worthy of publication after revisions -- but I'm not sure whether it's more suited to G3 vs. Genetics (or another more specialized journal).

One aspect where I'm a bit torn is the realism and value of different ways of modelling -- analytical approaches provide clearer understanding than simulations but are limited by their assumptions and scope. Thus, while I like the focus of the authors on the importance of an inversion capturing a locally adapted haplotype (which has been very poorly considered in previous analytical models), the restriction to a two-locus model given the analytical approach makes it hard to know how general these findings will be for cases where an inversion spans more than 2 causal loci. Simulation studies are better suited to this and indeed the recent work by Schaal et al. (2021) shows that inversions evolve more readily with strong selection in a polygenic context quite nicely. So is an analytical solution really that novel, especially it is restricted to just two loci? Other theory papers have also explored this area recently and gone beyond this paper in various ways (e.g. Proulx and Teotonio 2022; Connallon and Olito 2021).

That said, a strength of the approach here is that it helps clarify the results from some other models that have considered multiple loci in the continent-island model and are highly cited (Kirkpatrick and Barton 2006; Charlesworth and Barton 2018). Charlesworth and Barton 2018 assume that the focal (island) population is predominantly composed of individuals carrying a haplotype with all the locally favoured alleles -- which would not actually occur unless selection on each locus was really strong. This is an example of an analytically convenient but biologically unrealistic assumption, and it yields some counter-intuitive results in Charlesworth and Barton 2018, eq. 6C: $s_{\text{favouring_inversion}} = m * (n - 1) * r / (n * s)$, whereby $s_{\text{favouring_inversion}}$ decreases with increasing s (for tight linkage). Similarly, with their QLE approach, the prediction of $s_{\text{favouring_inversion}} = (n - 1) * m$ is still at odds with the results shown in Figure 3 here, which is biologically more realistic in its assumptions. These weird results come because calculating the rate at which an inversion invades is only looking at half the picture -- this assumption that the inversion would capture all of the locally beneficial alleles neglects the importance of selection in establishing the conditions for the assumption to hold -- as the chance of this holding would become very small with weaker selection and larger number of loci. This is nicely shown for the continent-island model in Figure 3 in the present paper, and I like how this paper has illustrated the importance of considering both invasion probability and the probability of capturing the right haplotype. I seem to remember that some inversion theory paper has covered this issue before using simulations but I haven't found which one -- i.e., showing that the number of locally-beneficial vs. locally-deleterious alleles that are captured in the inversion is critical. In any case, clarifying the literature on analytical models in this area seems worthwhile, and the paper the authors have submitted is clearly written but could be improved by more explicit discussion of these issues or especially, by extension to include more than two loci.

Overall, I think the main strength of this paper is in clarifying previous theoretical papers and setting straight some aspects of the literature that were a bit murky. I'm split as to whether I think it is more appropriate for G3 vs. Genetics, and have offered some suggestions below for how I think it's impact and usefulness could be improved.

Major comments:

1. I think more clarity is needed about what previous models show, what their assumptions are that lead to such predictions, and where they are wrong. I like the approach that leads to Figure 3 (combining invasion probability with haplotype frequency) and this should be compared with predictions from previous continent-island models (eg. line 295-296). It seems strange to emphasize that continent-island models didn't depend on selection when it was more the assumption that an inversion would capture both locally adapted loci that resulted in a lack of dependence on selection. For example, on line 4 in the abstract it states that continent-island models lead to higher probability of establishment when selection is weak -- but this is in fact not the case as shown in Figure 3 (max at intermediate selection for CI model). Care should be taken throughout the manuscript to really clarify the issues described above and make clear what is due to continent-island vs. two-patch model and what is due to focusing only on the rate of increase of an inversion when rare (invasion probability) vs. the combined probability of capturing favourable alleles + invading (also need to clarify on line 10, and other places throughout introduction/discussion).

Lots of other models have looked at inversions recently -- can you synthesize/compare their findings here? In reviewing this study I went back and looked at how some papers had cited the recent Charlesworth + Barton 2018 paper and noticed that citations very rarely talk about the importance of s , focusing instead on simply the well-accepted recombination suppression effect. Could you perhaps include some literature review showing if people are discussing predictions about inversion evolution in ways that need to be rectified?

2. Some more discussion or ideally analysis about capturing more than two loci would be useful. Compare to other papers -- going beyond the short section on lines 360-370 would really improve the paper, especially some analysis.

3. Please provide PDF of the supplementary materials for readers that don't have mathematica.

Minor comments:

Line 10: "stronger local selection increases the flow of maladaptive alleles and favours inversions". This would be more clear if it stated "strong local selection increases the allele frequency divergence thereby increasing the frequency of maladaptive alleles in migrants, favouring inversions"

Line 18: differentiation is less likely when migration rates are high (but some migration is needed for inversions to be beneficial)

Line 27: ubiquity seems to overstate the case. Prevalence would be more appropriate.

Line 61-64: The sentence "for example" doesn't naturally follow the previous one because the former is about probability of capturing a combination while the latter is about the benefit of a given inversion. Please clarify.

Line 51-52: Akerman and Burger (2014) provide one analytical dissection of migration-recombination-selection dynamics but not interpreted with respect to inversion invasion.

Lines 156-192: making links to other papers that have analysed similar models might be useful here.

Eq 13 + 14: Both appear to be approximations, were some squared terms neglected? This should be stated.

Figure 1: What is the second order approximation here? (same as for Eq. 13 and 14?) Note there is a ".." in the caption.

Line 295-296: It's REALLY important here to not just repeat orthodoxy! This sentence is only true for the invasion probability, not for the combined haplotype frequency + invasion probability, which is far more important, as shown in Figure 3. Please make sure to talk clearly about these components in the Discussion.

Line 301: and continent-island models still depend on selection, but in a somewhat more complicated way than implied by Kirkpatrick/Barton and Barton/Charlesworth papers.

Reviewer #2 (Comments for the Authors (Required)):

This paper analyses the invasion probability of an inversion (which completely suppresses recombination) in a two-locus, two-deme model, in which the two demes are connected via (potentially) asymmetric migration, and the two loci are partially linked and under divergent directional selection. The figures are based on numerical iteration of deterministic recursions for two biallelic loci, although there is also an attempt to furnish analytical QLE predictions. The paper thus attempts to generalise earlier deterministic analyses of the spread of inversions under continent-island migration (Kirkpatrick & Barton, 2006; Burger and Akerman 2011; Charlesworth & Barton 2018) and more recently in a two-deme model with perfectly symmetric migration (Proulx & Teotonio, 2022), by considering migration rates with arbitrary levels of asymmetry (thus bridging the continuum between the continent-island and perfectly symmetric migration limits). It is, however, less general in that it neglects other aspects such as dominance, epistasis, multiple loci etc. which are considered by some of these earlier models.

To the best of my understanding, the mathematical and numerical analysis in this paper is flawed in several ways (see detailed comments below): I thus focus on highlighting these, rather than on the findings of the study, which may change after a corrected analysis. However, I do think that it would be important to spend more effort (e.g., in the Discussion) in clarifying where models with asymmetric migration fall in reference to the two limiting cases (continent-island vs. two-deme) vis-a-vis conditions for establishment of an inversion: perhaps the QLE predictions, once corrected (they appear to be incorrect in their present form), can provide some intuition.

Detailed comments:

- Lines 94-95 The way migration is dealt with does not seem to be logically consistent. If a proportion m_{21} of juveniles in deme 1 are migrants from deme 2 (lines 94-95), then this implies that the proportion of non-migrants (i.e., juveniles with "resident" parents) should be $1 - m_{21}$. However, in the equations below, e.g., equation 1, the genotype frequency in deme 1 (for example) after migration is taken to be the weighted sum of genotype frequencies in demes 1 and 2 (before migration) with corresponding weights equal to $1 - m_{12}$ and m_{21} , rather than $1 - m_{21}$ and m_{21} . There seems to be a similar problem in equation 3.

For instance, if deme 2 is the continent and deme 1 the island, then setting $m_{12}=0$ in eq. 1 does not appear to give the correct equation for continent-island migration.

- Lines 139-145 and Equation 4: I don't think this is the right way to calculate the combined probability. Calculating the combined invasion probability boils down to modeling the initial spread of the inversion (arising in any one deme) as a two-type branching process where the two types correspond to the two demes. More precisely, one needs to explicitly calculate the probabilities u_1 and u_2 that a single inversion arising in deme 1 or deme 2 escapes extinction, and then write down $\gamma_i = u_1 \cdot (\text{freq of A1B1 in deme 1}) + u_2 \cdot (\text{freq of A1B1 in deme 2})$. This is straightforward using the standard theory of multitype branching process (see various early papers on fixation/establishment probabilities in subdivided populations by Pollak, Barton, Slatkin etc.). However, as far as I know, this does not boil down to equation 4.

Note that the condition $\lambda_{ii} > 1$ just means that there is a non-zero probability that an inversion arising anywhere (in the 2-deme population) will establish.

- The QLE predictions for allele frequencies (equations 6, 8 and 11) are incorrect (these appear to actually be the LE predictions), although the QLE predictions for the D itself (equations 7,9 and 12) appear to be correct. One can find the correct QLE predictions for arbitrary selection and migration in earlier work (see, e.g., eq. 3.21 of Akerman and Burger, 2014 for arbitrary migration, and eq. 4.3 of Burger and Akerman, 2011 for continent-island migration). In general, the QLE predictions for allele frequencies will depend also on r/s (not just m/s).

- Line 199 and equation 13: This isn't quite correct unless $s/r \rightarrow 0$, i.e., there is perfect LE (see equation 4 of Charlesworth & Barton, 2018). There was a mistake in the QLE analysis of Kirkpatrick & Barton 2006, which Charlesworth & Barton 2018 corrects. Related to this: it is incorrect to say that "increasing the strength of selection within the island has no effect to leading order" in small parameters (lines 203-204)- what matters are the ratios m/s and s/r ; see again eq. 4 of Charlesworth & Barton, 2018.

I suspect the QLE prediction for the growth rate of the inversion in the symmetric case (equation 14) will also involve s/r (once both the QLE analysis for the equilibrium allele frequencies is done correctly).

- I am skeptical whether the so-called "exact equilibrium calculated to second order to selection and migration" (which is the basis of all the numerical results and figures shown in this paper) is correct. This is because the computation of this exact equilibrium is based on the function "secondOrder" (in the Mathematica notebook) which is defined in the same kind of way as the function "linearise" which yields the erroneous QLE predictions. Since the plots/results are anyway based on numerics, why not numerically solve the exact recursions? These are quite simple, especially in the continuous time limit. As another check, note that there are explicit analytical predictions (which make no assumptions beyond the continuous time limit), e.g., equation A8 of Charlesworth and Barton 2018.

Associate Editor Comments:

Associate Editor Comments

Two experts in the field have reviewed your manuscript, and I have read it as well. While both reviewers (and myself) find that the general topic is of interest, reviewer 2 points to a number of important methodological issues (see reviewers comments below). In particular, some of the indices in equations 1-3 seem wrong (this may correspond to typos, but should be checked), and more importantly, the reasoning deriving invasion probabilities from "invasion fitness" (asymptotic rate of increase in the deterministic limit) is unsound, as invasion probability should mostly depend on events occurring during the first generations after the inversion has appeared in a given deme, before the asymptotic rate of increase has been reached. Along the same vein, the reasoning leading to eq. 4 is not justified and does not seem correct. As proposed by the reviewer, a multitype branching process model would probably be the most straightforward way of deriving invasion probabilities in this model. Furthermore, most of the results seem to be obtained from numerical analyses of 2d order approximations, and it is not clear why the authors do not simply perform numerical analyses of the exact model instead. Reviewer 2 also raises doubts on the validity of these 2d order approximations and of the QLE approximations presented in the paper.

While your manuscript is not currently acceptable for publication in GENETICS, we would welcome a substantially revised manuscript addressing the comments of both reviewers. We realize that addressing the methodological issues would imply of substantial amount of extra work, but this would also make the conclusions of the paper much stronger.

We look forward to receiving your revised manuscript. Please let the editorial office know approximately how long you expect to need for revisions.

Upon resubmission, please include:

1. A clean version of your manuscript;
2. A marked version of your manuscript in which you highlight significant revisions carried out in response to the major points raised by the editor/reviewers (track changes is acceptable if preferred);
3. A detailed response to the editor's/reviewers' feedback and to the concerns listed above. Please reference line numbers in this response to aid the editor and reviewers.

We have read through the thoughtful comments of the AE and two Reviewers and made changes to the paper following their advice, which we point out in detail below. Following recommendations from both the AE and Reviewer 2, we have substantially revised the methods used to derive the results. We note that this resulted in quantitative rather than qualitative changes, so that the results and conclusions from the first version of the manuscript still hold.

We apologise for the length of time it has taken to produce the revised MS. The main author, Carl Mackintosh, has been busy preparing his PhD thesis and applying for a postdoctoral position which delayed completion of the new version.

Reviewer #1 (Comments for the Authors (Required)):

The paper uses a well-studied model to show an intuitive but important clarification: that the benefit of an inversion establishing increases with the strength of selection on the locally adapted alleles. Lack of dependence on s is a weird quirk of previous analyses of continent-island model (and their assumptions to solve it) and it is conceptually important to show that other approaches to analysing such models arrive at different conclusions. Another paper has derived analytical machinery that could be used to lead to many of the same findings here (i.e. Akerman and Burger 2014) but it did not actually focus on inversions or evolution of recombination so the deployment of the model here is still novel and useful. Simulation studies have shown similar findings to those of this paper, but cannot be used to give rule-of-thumb predictions in the same way as analytical models. Given the prevalence of empirical studies on this topic and the murkiness of some of the previous theory papers, this paper is well worthy of publication after revisions -- but I'm not sure whether it's more suited to G3 vs. Genetics (or another more specialized journal).

Thank you for these positive comments.

One aspect where I'm a bit torn is the realism and value of different ways of modelling -- analytical approaches provide clearer understanding than simulations but are limited by their assumptions and scope. Thus, while I like the focus of the authors on the importance of an inversion capturing a locally adapted haplotype (which has been very poorly considered in previous analytical models), the restriction to a two-locus model given the analytical approach makes it hard to know how general these findings will be for cases where an inversion spans more than 2 causal loci. Simulation studies are better suited to this and indeed the recent work by Schaal et al. (2021) shows that inversions evolve more readily with strong selection in a polygenic context quite nicely. So is an

analytical solution really that novel, especially it is restricted to just two loci? Other theory papers have also explored this area recently and gone beyond this paper in various ways (e.g. Proulx and Teotonio 2022; Connallon and Olito 2021).

That said, a strength of the approach here is that it helps clarify the results from some other models that have considered multiple loci in the continent-island model and are highly cited (Kirkpatrick and Barton 2006; Charlesworth and Barton 2018). Charlesworth and Barton 2018 assume that the focal (island) population is predominantly composed of individuals carrying a haplotype with all the locally favoured alleles -- which would not actually occur unless selection on each locus was really strong. This is an example of an analytically convenient but biologically unrealistic assumption, and it yields some counter-intuitive results in Charlesworth and Barton 2018, eq. 6C: $s_{\text{favouring_inversion}} = m * (n - 1) * r / (n * s)$, whereby $s_{\text{favouring_inversion}}$ decreases with increasing s (for tight linkage). Similarly, with their QLE approach, the prediction of $s_{\text{favouring_inversion}} = (n-1) * m$ is still at odds with the results shown in Figure 3 here, which is biologically more realistic in its assumptions. These weird results come because calculating the rate at which an inversion invades is only looking at half the picture -- this assumption that the inversion would capture all of the locally beneficial alleles neglects the importance of selection in establishing the conditions for the assumption to hold -- as the chance of this holding would become very small with weaker selection and larger number of loci. This is nicely shown for the continent-island model in Figure 3 in the present paper, and I like how this paper has illustrated the importance of considering both invasion probability and the probability of capturing the right haplotype. I seem to remember that some inversion theory paper has covered this issue before using simulations but I haven't found which one -- i.e., showing that the number of locally-beneficial vs. locally-deleterious alleles that are captured in the inversion is critical. In any case, clarifying the literature on analytical models in this area seems worthwhile, and the paper the authors have submitted is clearly written but could be improved by more explicit discussion of these issues or especially, by extension to include more than two loci.

We combine our response to the two paragraphs above. We agree with Reviewer 1 and have added some extra comments in the Discussion on the limitation of the study to the capture of alleles at two segregating loci (paragraph starting line 444). Considering more than two loci would certainly be of interest and useful, but it dramatically increases the complexity of the model. A major consideration is that as the number of loci contributing to adaptation increases, the frequency of capture of a full set decreases. This may mean that an inversion which partially captures adaptive alleles results in the exclusion of those outside the inversion. We prefer to comment on this rather than analyse it, as that would require another paper-length project. We conclude "The relationship between invasion fitness and haplotype frequencies as the number of loci increases remains to be explored, but we expect inversion evolution will continue to depend on a balance between the selective advantage of the captured haplotype and on the probability of capturing a favourable haplotype."

Overall, I think the main strength of this paper is in clarifying previous theoretical papers and setting straight some aspects of the literature that were a bit murky. I'm split as to whether I think it is more appropriate for *Genetics* vs. *Genetics*, and have offered some suggestions below for how I think its impact and usefulness could be improved.

Thank you for your positive comments and suggestions, which have greatly improved our manuscript. We would like this paper to appear in *Genetics* for the simple reason that this is where the original papers by Kirkpatrick & Barton 2006 and Charlesworth & Barton 2018 were published. We believe this manuscript makes a significant contribution to the field by explicitly considering the conditions required for the formation of locally adaptive inversions, which have so far been neglected and are crucial to our understanding of how they can evolve. The paper has already been cited several times as a pre-print as it makes a major contribution to the theoretical understanding of inversion evolution. Empirical investigation of inversions has undergone a massive revival over the last few years due to the improvement and availability of long-read sequencing. We believe the paper will help understanding amongst this group of investigators and *Genetics* is the natural place where they will find this paper.

Major comments:

1. I think more clarity is needed about what previous models show, what their assumptions are that lead to such predictions, and where they are wrong. I like the approach that leads to Figure 3 (combining invasion probability with haplotype frequency) and this should be compared with predictions from previous continent-island models (eg. line 295-296). It seems strange to emphasize that continent-island models didn't depend on selection when it was more the assumption that an inversion would capture both locally adapted loci that resulted in a lack of dependence on selection. For example, on line 4 in the abstract it states that continent-island models lead to higher probability of establishment when selection is weak -- but this is in fact not the case as shown in Figure 3 (max at intermediate selection for CI model). Care should be taken throughout the manuscript to really clarify the issues described above and make clear what is due to continent-island vs. two-patch model and what is due to focusing only on the rate of increase of an inversion when rare (invasion probability) vs. the combined probability of capturing favourable alleles + invading (also need to clarify on line 10, and other places throughout

introduction/discussion).

Thank you for this observation. We have made extensive efforts throughout to clarify these issues, and have also clarified the terminology used in the manuscript, explicitly referring to "invasion" (given that an inversion exists), "capture" (which is the probability an inversion contains both adaptive alleles) and "establishment" (which is the joint probability of invasion and capture). See the Abstract for an example.

Lots of other models have looked at inversions recently -- can you synthesize/compare their findings here? In reviewing this study I went back and looked at how some papers had cited the recent Charlesworth + Barton 2018 paper and noticed that citations very rarely talk about the importance of s , focusing instead on simply the well-accepted recombination suppression effect. Could you perhaps include some literature review showing if people are discussing predictions about inversion evolution in ways that need to be rectified?

We agree there is a large literature. We have attempted to cover the theoretical literature fully in the Introduction.

2. Some more discussion or ideally analysis about capturing more than two loci would be useful. Compare to other papers -- going beyond the short section on lines 360-370 would really improve the paper, especially some analysis.

See answer above.

3. Please provide PDF of the supplementary materials for readers that don't have mathematica.

We apologise for not providing this in the first instance, and have done so this time.

Minor comments:

Thank you for the comments below, which we address individually:

Line 10: "stronger local selection increases the flow of maladaptive alleles and favours inversions". This would be more clear if it stated "strong local selection increases the allele frequency divergence thereby increasing the frequency of maladaptive alleles in migrants, favouring inversions"

Line 18: differentiation is less likely when migration rates are high (but some migration is needed for inversions to be beneficial)

Line 27: ubiquity seems to overstate the case. Prevalence would be more appropriate.

We have altered the manuscript as suggested for the three comments above.

Line 61-64: The sentence "for example" doesn't naturally follow the previous one because the former is about probability of capturing a combination while the latter is about the benefit of a given inversion. Please clarify.

This sentence has been made clearer.

Line 51-52: Akerman and Burger (2014) provide one analytical dissection of migration-recombination-selection dynamics but not interpreted with respect to inversion invasion.

This sentence now makes explicit reference to the dissection of these factors with respect to locally adaptive inversions, noting also that Proulx and Teotonio 2022 do so in terms of the genetic load present in the population.

Lines 156-192: making links to other papers that have analysed similar models might be useful here.

We have included references to papers analysing similar models throughout this section.

Eq 13 + 14: Both appear to be approximations, were some squared terms neglected? This should be stated.

We point out first that this analysis has been redone following feedback from another reviewer – and incorporating this feedback, we state the order to which we approximate in the revised version of the manuscript.

Figure 1: What is the second order approximation here? (same as for Eq. 13 and 14?) Note there is a "." in the caption.

This figure now shows an exact numerical solution.

Line 295-296: It's REALLY important here to not just repeat orthodoxy! This sentence is only true for the invasion

1st Revision - Authors' Response to Reviewers: January 27, 2024

probability, not for the combined haplotype frequency + invasion probability, which is far more important, as shown in Figure 3. Please make sure to talk clearly about these components in the Discussion.

Line 301: and continent-island models still depend on selection, but in a somewhat more complicated way than implied by Kirkpatrick/Barton and Barton/Charlesworth papers.

We have addressed both these comments through making sure that we are clear in our use of the terms "invasion" and "establishment", as mentioned earlier.

Reviewer #2 (Comments for the Authors (Required)):

This paper analyses the invasion probability of an inversion (which completely suppresses recombination) in a two-locus, two-deme model, in which the two demes are connected via (potentially) asymmetric migration, and the two loci are partially linked and under divergent directional selection. The figures are based on numerical iteration of deterministic recursions for two biallelic loci, although there is also an attempt to furnish analytical QLE predictions. The paper thus attempts to generalise earlier deterministic analyses of the spread of inversions under continent-island migration (Kirkpatrick & Barton, 2006; Buerger and Akerman 2011; Charlesworth & Barton 2018) and more recently in a two-deme model with perfectly symmetric migration (Proulx & Teotonio, 2022), by considering migration rates with arbitrary levels of asymmetry (thus bridging the continuum between the continent-island and perfectly symmetric migration limits). It is, however, less general in that it neglects other aspects such as dominance, epistasis, multiple loci etc. which are considered by some of these earlier models.

To the best of my understanding, the mathematical and numerical analysis in this paper is flawed in several ways (see detailed comments below): I thus focus on highlighting these, rather than on the findings of the study, which may change after a corrected analysis. However, I do think that it would be important to spend more effort (e.g., in the Discussion) in clarifying where models with asymmetric migration fall in reference to the two limiting cases (continent-island vs. two-deme) vis-a-vis conditions for establishment of an inversion: perhaps the QLE predictions, once corrected (they appear to be incorrect in their present form), can provide some intuition.

Thank you for these comments which we address individually below.

Detailed comments:

- Lines 94-95 The way migration is dealt with does not seem to be logically consistent. If a proportion m_{21} of juveniles in deme 1 are migrants from deme 2 (lines 94-95), then this implies that the proportion of non-migrants (i.e., juveniles with "resident" parents) should be $1-m_{21}$. However, in the equations below, e.g., equation 1, the genotype frequency in deme 1 (for example) after migration is taken to be the weighted sum of genotype frequencies in demes 1 and 2 (before migration) with corresponding weights equal to $1-m_{12}$ and m_{21} , rather than $1-m_{21}$ and m_{21} . There seems to be a similar problem in equation 3. For instance, if deme 2 is the continent and deme 1 the island, then setting $m_{12}=0$ in eq. 1 does not appear to give the correct equation for continent-island migration.

Thank you for pointing this out – it has now been altered and is consistent.

- Lines 139-145 and Equation 4: I don't think this is the right way to calculate the combined probability. Calculating the combined invasion probability boils down to modeling the initial spread of the inversion (arising in any one deme) as a two-type branching process where the two types correspond to the two demes. More precisely, one needs to explicitly calculate the probabilities u_1 and u_2 that a single inversion arising in deme 1 or deme 2 escapes extinction, and then write down $\gamma_i = u_1 * (\text{freq of A1B1 in deme 1}) + u_2 * (\text{freq of A1B1 in deme 2})$. This is straightforward using the standard theory of multitype branching process (see various early papers on fixation/establishment probabilities in subdivided populations by Pollak, Barton, Slatkin etc.). However, as far as I know, this does not boil down to equation 4. Note that the condition $\lambda_{ii} > 1$ just means that there is a non-zero probability that an inversion arising anywhere (in the 2-deme population) will establish.

Thank you for this suggestion. We now plot invasion probabilities calculated using a multitype branching process. While the eigenvalue does not strictly give the fixation probability, it still corresponds to a rate of growth within the metapopulation. Therefore, we decided to continue to analyse it because the forces that increase the eigenvalue are likely to increase the fixation probability, even if they are not quantitatively the same.

- The QLE predictions for allele frequencies (equations 6, 8 and 11) are incorrect (these appear to actually be the LE predictions), although the QLE predictions for the D itself (equations 7,9 and 12) appear to be correct. One can find the correct QLE predictions for arbitrary selection and migration in earlier work (see, e.g., eq. 3.21 of Akerman and Burger, 2014 for arbitrary migration, and eq. 4.3 of Burger and Akerman, 2011 for continent-island migration). In general, the QLE predictions for allele frequencies will depend also on r/s (not just m/s).

1st Revision - Authors' Response to Reviewers: January 27, 2024

- Line 199 and equation 13: This isn't quite correct unless $s/r \rightarrow 0$, i.e., there is perfect LE (see equation 4 of Charlesworth&Barton, 2018). There was a mistake in the QLE analysis of Kirkpatrick & Barton 2006, which Charlesworth & Barton 2018 corrects. Related to this: it is incorrect to say that "increasing the strength of selection within the island has no effect to leading order" in small parameters (lines 203-204)- what matters are the ratios m/s and s/r ; see again eq. 4 of Charlesworth&Barton, 2018.

I suspect the QLE prediction for the growth rate of the inversion in the symmetric case (equation 14) will also involve s/r (once both the QLE analysis for the equilibrium allele frequencies is done correctly).

Thank you for pointing this out. You are correct in that the expressions we derived were not QLE predictions, but rather LE predictions. This was due to a mistake in the method that led to the $O(1/r)$ terms vanishing for the allele frequencies. Following your advice, we (re)derive the QLE allele frequencies such that they match those of Akerman & Bürger/Bürger & Akerman and obtain an expression for the eigenvalue in the symmetric case.

- I am skeptical whether the so-called "exact equilibrium calculated to second order to selection and migration" (which is the basis of all the numerical results and figures shown in this paper) is correct. This is because the computation of this exact equilibrium is based on the function "secondOrder" (in the Mathematica notebook) which is defined in the same kind of way as the function "linearise" which yields the erroneous QLE predictions. Since the plots/results are anyway based on numerics, why not numerically solve the exact recursions? These are quite simple, especially in the continuous time limit. As another check, note that there are explicit analytical predictions (which make no assumptions beyond the continuous time limit), e.g., equation A8 of Charlesworth and Barton 2018.

As discussed above, the error was in the method rather than in the Mathematica functions. We now numerically solve the recursions exactly, as numerical methods are required to find the invasion probabilities from the branching process.

February 27, 2024

GENETICS-2024-306820

The establishment of locally adaptive inversions in structured populations

Dear Dr. Pomiankowski:

Two experts in the field have reviewed your manuscript, and I have read it as well. I am pleased to inform you that, with minor revisions, it is potentially suitable for publication in GENETICS. Both reviewers see that this MS is a substantial improvement, and that it addresses most of the problems with the previous version. However, the reviewers (especially rev. 2) have comments and concerns that must be addressed in a revised manuscript. You can read their reviews at the end of this email.

It is most important that you address the apparent mistakes picked up by rev. 2, but also please explore the results with lower recombination rates, and if possible, find the fixation probabilities in the continuous-time limit, as suggested.

We look forward to receiving your revised manuscript. Please let the editorial office know approximately how long you expect to need for revisions.

Upon resubmission, please include:

1. A clean version of your manuscript;
2. A marked version of your manuscript in which you highlight significant revisions carried out in response to the major points raised by the editor/reviewers (track changes is acceptable if preferred);
3. A detailed response to the editor's/reviewers' comments and to the concerns listed above. Please reference line numbers in this response to aid the editors.

Additionally, please ensure that your resubmission is formatted for GENETICS.

<https://academic.oup.com/genetics/pages/general-instructions>

Follow this link to submit the revised manuscript: Link Not Available

Sincerely,

Nick Barton
Senior Editor
GENETICS

Approved by:
Howard Lipshitz
Editor in Chief
GENETICS

Reviewer #1 (Comments for the Authors (Required)):

The paper is much improved in this revision -- the authors have responded to my previous points appropriately. I had not caught the problems with the methods identified by the editor and the other reviewer, but the new version appears to fix these problems. Some of the explanation about the rationale for methodological choices seems a little unclear to me (e.g. line 174; comment below). I checked a few cases and the $2^*(\lambda-1)$ approximation for establishment probability matches pretty close to the branching process solution (see attached, red thick lines correspond to $2^*(\lambda-1)$ from Eq. 14, with $r = 0.5$), so it seems worth at least showing this comparison somewhere in the paper or supp mat. Without actually showing the $2^*(\lambda-1)$ approximation, the discussions about why it might not work well seem out of place/unnecessary. Furthermore, if Kimura's equation is used instead of Haldane's 2^* s with the asymptotic approximation, it can potentially account for small population size. I would be curious to see how well the predictions of the various numerical solutions and approximations match results from individual-based Wright-Fisher simulations (e.g. with SLiM), particularly at smaller population sizes which might cause some deviations.

Specific comments:

- line 174: It's not clear to me why migration will have a particularly strong effect on the accuracy of the asymptotic

approximation to establishment probability when the inversion is rare for the case with two demes and bi-directional migration. Isn't the chance of the inversion migrating to the other deme and then migrating back quite rare, when the inversion itself is rare? And doesn't the asymptotic approximation still account for bi-directional migration? Does any problem relative to the branching process approach not arise from approximations involved here? The agreement between $2^*(\lambda-1)$ and the branching process seems pretty good in a few cases I checked. I maybe missing the point, but this could be better explained.

- Line 278: effect LD within

- line 173: needs to to

Reviewer #2 (Comments for the Authors (Required)):

The authors address the technical points raised in the previous review and also clarify the differences between asymmetric and symmetric migration more explicitly. However, I still have some comments and suggestions.

Figures 1 and 3: $r=0.1$ and $r=0.5$ are perhaps unrealistically high recombination rates. In particular, $r=0.5$ corresponds to essentially unlinked loci, and it seem very unlikely that an inversion would span such large genomic distances. Also, I am curious whether the slightly non-monotonic dependence of invasion probabilities on s (that one observes in figure 1 and 3 for $r=0.1$ in the symmetric scenario) becomes more pronounced as one goes to smaller, i.e., more realistic values of r , say $r=0.01$. This also relates to the general conclusion stated in lines 478-480: Are inversions still "particularly likely to arise and establish when selection on locally adaptive alleles is strong" if one considers more realistic values of r , or would invasion probabilities then be maximum for intermediate values of s ? Maybe some insight can also be gained from the explicit expressions in equations 13 and 14.

Technical points:

- Equations 1 and 2 still appear to be inconsistent with the definition of migration rates in lines 141 and 142. If "a proportion $m_{\{kl\}}$ of juveniles in deme l are migrants from deme k ", then equations 1 and 2, which describe genotype frequencies in deme k should involve the migration rates $m_{\{lk\}}$ (and not $m_{\{kl\}}$). I think Equation 3 itself is correct, though.

- It may be possible to obtain explicit expressions for u_1 and u_2 (the survival probability of an inversion arising in either deme 1 or 2) in the continuous time limit. This basically involves Taylor expanding equation 17 in terms of "small parameters" i.e., s_1 , s_2 , m_{12} , m_{21} , u_1 and u_2 , and then retaining only terms that are quadratic in small parameters, e.g., u_1^2 or s_1^2 but not $u_1^2 s_1$.

- Equations 6 and 7 still appear to be incorrect. In analogy with equation 3.5 of Burger & Akerman 2014, $f_{\{0\}^{\{1\}}$ and $f_{\{0\}^{\{2\}}$ should represent the frequency of one of the alleles (B_1 or B_2) in deme 1 and deme 2, assuming linkage equilibrium. These are not "the frequencies of adaptive and non-adaptive alleles in deme 1" as stated in line 203. Also, $f_{\{0\}^{\{2\}}$ should then be $(1+2\alpha_2 - \dots)/2$, and not $(1+2\alpha_1 - \dots)/2$. I am not sure if this is just a typo or it ends up affecting QLE numerics that build upon equations 6 and 7.

Minor points:

Lines 289-290: It would be more accurate to say that high rates of migration override selection.

If the $f_{\{0\}^{\{1\}}$ and $f_{\{0\}^{\{2\}}$ in equation 7 are the same as those in equation 6, then they should have a \wedge symbol?

Equations 13 and 14: Maybe worth saying that s_{inv} just the leading value of the matrix in equation 3 minus 1?

Line 291: I find the sentence "This advantage .. nature of migrant gene flow" rather cryptic and hard to follow. Maybe rephrase? Isn't the basic idea that with increasing selection, selection is pretty efficient in eliminating migrant genotypes regardless of recombination rates?

Line 298: "similarly homogeneous for maladaptive alleles" is also cryptic: I don't think homogeneous is the right word.

Figure 2: I find the legend a bit confusing, as it takes a while to figure out what is happening. Maybe provide more details?

Dear Dr. Pomiankowski:

Two experts in the field have reviewed your manuscript, and I have read it as well. I am pleased to inform you that, with minor revisions, it is potentially suitable for publication in GENETICS. Both reviewers see that this MS is a substantial improvement, and that it addresses most of the problems with the previous version. However, the reviewers (especially rev. 2) have comments and concerns that must be addressed in a revised manuscript. You can read their reviews at the end of this email.

It is most important that you address the apparent mistakes picked up by rev. 2, but also please explore the results with lower recombination rates, and if possible, find the fixation probabilities in the continuous-time limit, as suggested.

We look forward to receiving your revised manuscript. Please let the editorial office know approximately how long you expect to need for revisions.

Upon resubmission, please include:

1. A clean version of your manuscript;
2. A marked version of your manuscript in which you highlight significant revisions carried out in response to the major points raised by the editor/reviewers (track changes is acceptable if preferred);
3. A detailed response to the editor's/reviewers' comments and to the concerns listed above. Please reference line numbers in this response to aid the editors.

Additionally, please ensure that your resubmission is formatted for GENETICS.

<https://academic.oup.com/genetics/pages/general-instructions>

Follow this link to submit the revised manuscript: Link Not Available

Sincerely,

Nick Barton
Senior Editor
GENETICS

Approved by:
Howard Lipshitz
Editor in Chief
GENETICS

Reviewer #1 (Comments for the Authors (Required)):

The paper is much improved in this revision -- the authors have responded to my previous points appropriately. I had not caught the problems with the methods identified by the editor and the other reviewer, but the new version appears to fix these problems. Some of the explanation about the rationale for methodological

choices seems a little unclear to me (e.g. line 174; comment below). I checked a few cases and the $2^*(\lambda-1)$ approximation for establishment probability matches pretty close to the branching process solution (see attached, red thick lines correspond to $2^*(\lambda-1)$ from Eq. 14, with $r = 0.5$), so it seems worth at least showing this comparison somewhere in the paper or supp mat. Without actually showing the $2^*(\lambda-1)$ approximation, the discussions about why it might not work well seem out of place/unnecessary.

It's true that in many cases the $2^*(\lambda-1)$ approximation works well, particularly when rates of migration are low. However, as pointed out by Reviewer 2 and the editor in a previous review, this is an asymptotic result. The branching process captures the effects of early extinction brought about by migration between demes. Once early extinction becomes unlikely, then the inversion's trajectory should follow the asymptotic rate of increase given by $\lambda - 1$. We found that the biggest discrepancy between the two methods was when migration rates were relatively high, exacerbating the possibility of early extinction relative to asymptotic growth.

To address your points, we have included a clearer justification for the method in the manuscript (lines 167-178) and provide a supplementary figure showing the divergence in invasion probabilities with higher rates of migration (fig S1). In this figure, we use the exact numerically calculated eigenvalue rather than the $O(1/r)$ approximation, since the branching process is also exact.

Furthermore, if Kimura's equation is used instead of Haldane's 2^* s with the asymptotic approximation, it can potentially account for small population size. I would be curious to see how well the predictions of the various numerical solutions and approximations match results from individual-based Wright-Fisher simulations (e.g. with SLiM), particularly at smaller population sizes which might cause some deviations.

The difference between the approximations is not a result of population size effects -- in fact, since our model has an infinite population size, Haldane's approximation should be accurate. Population size effects are not a focus of this manuscript. Variation in population size would only serve to add noise to the results through drift.

Specific comments:

- line 174: It's not clear to me why migration will have a particularly strong effect on the accuracy of the asymptotic approximation to establishment probability when the inversion is rare for the case with two demes and bi-directional migration. Isn't the chance of the inversion migrating to the other deme and then migrating back quite rare, when the inversion itself is rare? And doesn't the asymptotic approximation still account for bi-directional migration? Does any problem relative to the branching process approach not arise from approximations involved here? The agreement between $2^*(\lambda-1)$ and the branching process seems pretty good in a few cases I checked. I maybe missing the point, but this could be better explained.

We address this point in the comment above and have included an explanation in the text.

- Line 278: effect <of> LD within

- line 173: needs to to

Both of these typos have been corrected – thank you.

Reviewer #2 (Comments for the Authors (Required)):

The authors address the technical points raised in the previous review and also clarify the differences between asymmetric and symmetric migration more explicitly. However, I still have some comments and suggestions.

Figures 1 and 3: $r=0.1$ and $r=0.5$ are perhaps unrealistically high recombination rates. In particular, $r=0.5$ corresponds to essentially unlinked loci, and it seems very unlikely that an inversion would span such large genomic distances. Also, I am curious whether the slightly non-monotonic dependence of invasion probabilities on s (that one observes in figure 1 and 3 for $r=0.1$ in the symmetric scenario) becomes more pronounced as one goes to smaller, i.e., more realistic values of r , say $r=0.01$. This also relates to the general conclusion stated in lines 478-480: Are inversions still "particularly likely to arise and establish when selection on locally adaptive alleles is strong" if one considers more realistic values of r , or would invasion probabilities then be maximum for intermediate values of s ? Maybe some insight can also be gained from the explicit expressions in equations 13 and 14.

We agree $r = 0.5$ is an unrealistically high recombination rate. Following your advice and that of the Editor, we now present figures (1 and 3) where $r = 0.01, 0.05, \text{ or } 0.15$. The new figures show that with moderate recombination and migration rates ($r \geq 0.05, m \geq 0.01$), there is an increase in both invasion probability and establishment probability when local selection gets stronger. Importantly, the result for the establishment probability holds for both the continent-island and two-deme scenarios.

Stronger selection disfavours the invasion of inversions by generating LD between locally adaptive alleles (so that inversions are not necessary) but favours them by increasing the rate of maladaptive gene flow in the two-deme scenario. When recombination rates are low the LD built by selection is maintained, so that the former effect of selection dominates the latter. In this case, we see a relationship more like the continent-island model.

One example of inversion sizes can be found in Le Moan et al 2023 (on bioRxiv) where inversions span genetic distances of 1-40cM. From endpoint to endpoint, this would correspond to between locus recombination rates of $\sim 0.01-0.3$ (mostly $>5\text{cM} \sim r = 0.05$), though of course the loci involved are unlikely to be exactly at the

endpoints. So, we consider all the recombination rates used in the plots now to be realistic.

We would not expect small inversions ($r = 0.01$) to make large contributions to local adaptation through linking locally adaptive alleles because inversions are less advantageous when LD is already being maintained. This idea is supported by the above data --- though there could be observation bias in play, since smaller inversions are harder to detect.

As well as presenting figures with more realistic recombination rates, there is now a paragraph (lines 395-405) in the discussion on recombination rates, and we have adjusted the claims in the paper appropriately.

Technical points:

- Equations 1 and 2 still appear to be inconsistent with the definition of migration rates in lines 141 and 142. If "a proportion m_{kl} of juveniles in deme l are migrants from deme k", then equations 1 and 2, which describe genotype frequencies in deme k should involve the migration rates m_{lk} (and not m_{kl}). I think Equation 3 itself is correct, though.

This has now been corrected, thanks.

- It may be possible to obtain explicit expressions for u_1 and u_2 (the survival probability of an inversion arising in either deme 1 or 2) in the continuous time limit. This basically involves Taylor expanding equation 17 in terms of "small parameters" i.e., s_1 , s_2 , m_{12} , m_{21} , u_1 and u_2 , and then retaining only terms that are quadratic in small parameters, e.g., u_1^2 or s_1^2 but not $u_1^2 s_1$.

We attempted this; however we were unable to obtain any results that could be meaningfully interpreted. What we could get, we have put in the supplementary Mathematica file.

- Equations 6 and 7 still appear to be incorrect. In analogy with equation 3.5 of Burger & Akerman 2014, f_1 and f_2 should represent the frequency of one of the alleles (B1 or B2) in deme 1 and deme 2, assuming linkage equilibrium. These are not "the frequencies of adaptive and non-adaptive alleles in deme 1" as stated in line 203. Also, f_2 should then be $(1+2\alpha_2 - \dots)/2$, and not $(1+2\alpha_1 - \dots)/2$. I am not sure if this is just a typo or it ends up affecting QLE numerics that build upon equations 6 and 7.

This has now been corrected, and we thank the reviewer for their attention to detail.

Minor points:

Lines 289-290: It would be more accurate to say that high rates of migration override selection.

The sentence

“ This happens because selection reinforces local adaptation and makes migrants more maladapted, increasing the advantage of having the fittest genotype.”

has been rewritten as

“This happens because selection reinforces local adaptation so migrants bring more maladaptive alleles into the focal deme, which can increase the rate at which recombination degenerates LD between coadapted alleles.”

If the f_{0}^{1} and f_{0}^{2} in equation 7 are the same as those in equation 6, then they should have a \wedge symbol?

Yes, we have corrected this.

Equations 13 and 14: Maybe worth saying that s_{inv} just the leading value of the matrix in equation 3 minus 1?

We have added a line stating this.

Line 291: I find the sentence "This advantage .. nature of migrant gene flow" rather cryptic and hard to follow. Maybe rephrase? Isn't the basic idea that with increasing selection, selection is pretty efficient in eliminating migrant genotypes regardless of recombination rates?

This line has been rephrased.

Line 298: "similarly homogeneous for maladaptive alleles" is also cryptic: I don't think homogeneous is the right word.

This line has also been rephrased.

Figure 2: I find the legend a bit confusing, as it takes a while to figure out what is happening. Maybe provide more details?

We have amended the figure legend so that is clear what value the parameters take in each panel.

April 16, 2024

RE: GENETICS-2024-306996

Prof. Andrew Pomiankowski
University College London
Genetics, Evolution and Environment
Gower Street
London, N/A WC1E 6BT
United Kingdom

Dear Dr. Pomiankowski:

Congratulations! We are delighted to inform you that your manuscript entitled "The establishment of locally adaptive inversions in structured populations" is acceptable for publication in GENETICS. Many thanks for submitting your research to the journal. The MS deals with all the reviewers' comments well.

To Proceed to Production:

1. Format your article according to GENETICS style, as discussed at <https://academic.oup.com/genetics/pages/general-instructions>, and upload your final files at <https://genetics.msubmit.net>.
2. Your manuscript will be published as-is (unedited-as submitted, reviewed, and accepted) at the GENETICS website as an Advanced Access article and deposited into PubMed shortly after receipt of source files and the completed license to publish. Please notify sourcefiles@thegsajournals.org if you do not wish to publish your article via Advanced Access.
3. We invite you to submit an original color figure related to your paper for consideration as cover art. Please email your submission to the editorial office or upload it with your final files. You can submit a small-sized image for evaluation, and if selected, the final image must be a TIFF file 2513px wide by 3263px high (8.375 by 10.875 inches; resolution of 600ppi). Please avoid graphs and small type.

If you have any questions or encounter any problems while uploading your accepted manuscript files, please email the editorial office at sourcefiles@thegsajournals.org.

Sincerely,

Nick Barton
Senior Editor
GENETICS

Approved by:
Howard Lipshitz
Editor in Chief
GENETICS

note: Please add jnls.author.support@oup.com and genetics.oup@kwglobal.com (or the domains @oup.com and @kwglobal.com) to your email program's "safe senders" list. You will be contacted by both at various points during the production process.